# Occurrence of Hydroxytyrosol, Tyrosol and Their Metabolites in Italian Cheese

**DOI:** 10.3390/molecules28176204

**Published:** 2023-08-23

**Authors:** Danilo Giusepponi, Carolina Barola, Elisabetta Bucaletti, Simone Moretti, Fabiola Paoletti, Andrea Valiani, Raffaella Branciari, Roberta Galarini

**Affiliations:** 1Istituto Zooprofilattico Sperimentale dell’Umbria e delle Marche “Togo Rosati”, Via G. Salvemini 1, 06126 Perugia, Italy; d.giusepponi@izsum.it (D.G.); c.barola@izsum.it (C.B.); e.bucaletti@izsum.it (E.B.); s.moretti@izsum.it (S.M.); f.paoletti@izsum.it (F.P.); a.valiani@izsum.it (A.V.); 2Department of Veterinary Medicine, University of Perugia, Via San Costanzo 4, 06126 Perugia, Italy; raffaella.branciari@unipg.it

**Keywords:** cheese, hydroxytyrosol, tyrosol, sulphated metabolites, glucuronate metabolites, LC-Q-Orbitrap

## Abstract

Tyrosol (T) and hydroxytyrosol (HT) are phenyl alcohol polyphenols with well-recognized health-promoting properties. They are widely diffused in several vegetables, especially in olive products (leaves, fruits and oil). Therefore, they could be present in food produced from herbivorous animals such as in milk and cheese. In this study, an analytical method to determine T, HT and some of their phase II metabolites (sulphates and glucuronides) in cheese was developed and validated. Samples were extracted with an acidic mixture of MeOH/water 80/20 (*v*/*v*) and, after a low temperature clean-up, the extracts were evaporated and injected in a liquid-chromatography coupled with high resolution mass spectrometry (LC-Q-Orbitrap). A validation study demonstrated satisfactory method performance characteristics (selectivity, linearity, precision, recovery factors, detection and quantification limits). The developed protocol was then applied to analyze 36 Italian cheeses made from ewe, goat and cow milk. The sum of detected compounds (T, tyrosol sulfate, hydroxytyrosol-3-O-sulfate and hydroxytyrosol-4-O-sulfate) reached as high as 2300 µg kg^−1^ on a dry weight basis, although in about 45% of cow cheeses it did not exceed 50 µg kg^−1^. Ewe cheeses were significantly richer of polyphenols (sum) as well as HT sulfate metabolites than cow cheeses. In conclusion, results shows that cheese cannot be considered an important dietary source of these valuable compounds.

## 1. Introduction

There is extensive literature on the presence of polyphenols belonging to phenyl alcohol sub-class, tyrosol (T) and hydroxytyrosol (HT), in various plants and, mainly, in olive products (fruits, leaves and oil) as these molecules have demonstrated several beneficial health effects. Research has clearly shown that the consumption of HT contributes to the protection of blood lipids from oxidative stress such that to be inserted in the list of health claims of EU Commission Regulation 432/2012 [1]. A daily intake of at least five milligrams of HT and T (20 g of olive oil) is required to obtain this beneficial effect. Moreover, there is evidence that HT and T offer protection against the development of cancer, cardiovascular diseases, type 2 diabetes, obesity and metabolic syndrome. Although extensively studied, the exact molecular mechanisms underlying many of these favorable effects are yet to be fully understood. Certainly, a wide variety of HT and T biological activities is associated with antioxidant action due to their chemical features, which mainly prevent the formation of free radicals formed during autoxidation processes [2].

The olive oil industry involves the generation of a large amount of solid (pomace) and liquid (mill wastewater) by-products containing high concentrations of HT, T and more complex polyphenols. These wastes have a great impact on terrestrial and aquatic environment due to their high phytotoxicity and, in the Mediterranean area, there is great interest about their disposal. Many efforts are also being made to reutilize them in feed industry [3]. Among their beneficial effects, polyphenols also have antimicrobial activity [4] and, when supplemented to animal feed, have shown the potential to reduce the use of veterinary drugs in farming [5]. There are several studies measuring HT, T and some of their phase II metabolites in biological fluids of humans and laboratory animals (rats) after treatment with phenol-enriched food and feed, but very few are papers reporting methods and data in tissues and body fluids of food-producing animals. However, when olive oil by-products are supplemented to animal feed, the evaluation of polyphenol concentrations in derived food (meat, milk, eggs and cheese) can contribute to fully understand the observed beneficial effects on nutritional, microbiological and technological food quality properties. In this context, in our previous research we have measured HT, T and some of their phase II conjugate metabolites (sulphates and glucuronates) in poultry and rabbit meat, ewe milk and cheese produced from animals fed diet-supplemented polyphenols from olive oil waste [6,7,8]. Being hydrophilic compounds, it was shown that only few parts per billions were sporadically detectable in tissues, but concentrations in the order of hundreds of part per billion were found both in the milk and cheese of ewes to which feed enriched with by-products of olive oil industry had been administered [7]. Since polyphenol compounds strongly interact with milk casein whereas little interaction was observed with whey proteins [9], it was not surprising that they passed directly from milk to cheese. Interestingly, in the same study Branciari and coworkers measured a mean of about 600 µg kg^−1^ dw of T also in the cheese of ewes belonging to the control group [7]. HT and T are contained in glycosylated form in several plants and, therefore, after hydrolysis, these compounds could also be found in food produced by farm animals fed with conventional feed. In addition, it has been documented that small amounts of HT and T could also derive from tyramine and dopamine metabolism. In particular, HT is a product of dopamine oxidative metabolism known as DOPET (3,4-dihydroxyphenylethanol) [10].

There is a great interest in nutraceutical and functional foods with remarkable antioxidant properties. Accordingly, several studies investigated the total polyphenol content (or, better, antioxidant capacity) of milk and cheese by means of spectrophotometric assays such as Folin-Ciocalteu, DPPH (2,2-diphenyl-1-picryl-hydrazyl-hydrate) and ABTS (2,2′-azino-bis [3-ethylbenzthiazoline-6-sulphonic acid]) [11,12,13,14,15,16]. This research assessed, among other factors, the influence of animal species (ewe, goat, cow and buffalo) and husbandry practices. Needing more complex and expensive instrumentation, less data are available on specific contents of polyphenol sub-classes obtained by means of more selective instrumental techniques such as liquid chromatography coupled with diode array detection (LC-DAD) or, better, liquid chromatography coupled with tandem mass spectrometry (LC-MS/MS) [17]. Therefore, frequently it is not exactly clear to which substances the antioxidant action is attributable.

The goal of this study was to develop and validate an analytical procedure for measuring T, HT and some of their phase II metabolites (glucuronides and sulfates) in cheese using liquid chromatography combined with high-resolution tandem mass spectrometry (LC-HRMS/MS). The method could be useful in investigations looking into the cause–effect relationships between polyphenol contents and health-promoting characteristics naturally present in food or gained after fortification with polyphenol-rich additives (functional foods). Finally, method applicability was proven by testing 36 commercial ewe, goat and cow cheeses.

## 2. Results

### 2.1. Method Development

Cheese is a very complex matrix, especially for its high fat content. To extract and purify cheese before the analysis of substances present in traces, two main approaches have been developed: defatting using liquid-liquid extraction with *n*-hexane and low-temperature clean-up, also called freezing-out [18,19,20,21,22,23]. During sample preparation optimization, satisfactory results were obtained adopting freezing-out (20 min at −80 °C) after each extraction. Freezing-out also had the advantage of not requiring operator supervision and reagent consuming. Subsequently, the introduction of an additional purification step through SPE cartridges was attempted. Initially, Phree^TM^ tubes produced specifically to remove phospholipids were used. Phospholipids are an important bioactive lipid class of milk and, from an analytical point of view, are insidious substances since they cannot be removed by classical degreasing techniques given their amphiphilic properties [24]. Experiments introducing a clean-up step with a Phree^TM^ cartridge after the freezing out gave unsatisfactory recovery factors of sulfate metabolites without any significant improvement. Similar results were obtained for two other kinds of SPE cartridges widely applied in analytical methods for traces, i.e., OASIS HLB Prime and OASIS HLB. As an example, the detrimental effect of purification with OASIS HLB cartridges on analyte recovery factors is shown in Figure 1. Sulphates are also negatively charged at an acidic pH, hampering their full retention on OASIS HLB polymer based on a reversed-phase retention mechanism. Therefore, the SPE clean up was excluded from the final protocol.

With regard to chromatographic separation, the most applied LC column is the Acquity BEH C18 produced by the Waters Corporation. Since the mid-2000s, Spanish researchers have conducted pioneering studies about the metabolic fate of polyphenol substances of olive oil. Probably, the first paper describing the separation and tentative identification of conjugated metabolites of HT and T through liquid chromatography coupled to tandem mass spectrometry (LC-MS/MS) was published in 2005 by researchers of Madrid University [25]. Analyte separation was achieved using a 5-µm particle size Nucleosil 120 RP-18 column (250 mm × 4.6 mm, Machery Nagel, Düren, Germany) with 1% formic acid and acetonitrile as mobile phases. In 2006, de la Torre-Carbot and coworkers of Barcelona University proposed the use of a 3-µm particle size column (Luna C18, Phenomenex) with 0.1% formic acid and acetonitrile as eluents to separate HT, T, homovanillic acid and some of their metabolites in human low-density lipoproteins [26].

Finally, in 2009, the scientists of Lleida University described for the first time the application of a reversed-phase packing material with a particle size lower than 2-μm, i.e., UPLC Acquity BEH C18, in order to separate polyphenols and metabolites in human plasma using as mobile phases 0.2% acetic acid and acetonitrile [27]. Later, the majority of published papers adopted this chromatographic approach, also introducing different compositions of mobile phase (ammonium formate or formic acid instead acetic acid; MeOH instead of acetonitrile) [28,29,30,31,32,33,34,35].

That said, during method development the Acquity BEH C18 column was specifically chosen, as it focuses chromatographic optimization only on the eluent composition to achieve satisfactory separation even between the critical pair, HT-3-S and HT-4-S. Aqueous solutions of formic acid, acetic acid and ammonium formate were tested both with MeOH and with acetonitrile. The better results were obtained using ammonium formate (2 mM) and MeOH. The chromatograms and MS^2^ spectra recorded injecting a standard solution (300 ng mL^−1^) of the eight analytes are shown in Appendix A. It is worth noting that until 2013, metabolite analyses were only qualitative due to the lack of pure standards, but starting from that year many reference materials became commercially available, allowing quantitative measures and distinction among isomeric metabolites [32].

### 2.2. Method Validation

Method selectivity was achieved by acquisition of the molecular ion in addition to two fragment ions within a mass error of 5 ppm (or 1 mDa) and by checking the relative retention times. The linearity of matrix-matched calibration curves were ascertained in the range 0.5–100 ng mL^−1^ for TG, HT-3-S, HT-4-S, T-S, HT-3-G and HT-S, 0.5–1000 µg kg^−1^ for HT and 5–1000 µg kg^−1^ for T. The results of the validation study are detailed in Table 1. Except for HT-S at the first validation level (15 µg kg^−1^ dw), coefficients of variation observed in repeatability (CV_r_) and intra-laboratory reproducibility conditions (CV_wR_) were lower than 15% and 20%, respectively. Recovery factors were higher than 60% with an improvement for the more lipophilic analytes, HT and T (>70%). The limits of quantification (LOQs) were fixed at the first validation concentration, i.e., 15 µg kg^−1^ dw for all the analytes except for T (150 µg kg^−1^ dw). Accordingly, limits of detection (LODs) were fixed as one third of LOQs (5 µg kg^−1^ dw for all the analytes; 50 µg kg^−1^ dw for T).

### 2.3. Analysis of Real Samples

The quantification of polyphenol contents in thirty-six commercial cheese samples was carried out, preparing, within each analytical batch, a matrix-matched standard curve after subtraction of the relevant background area of incurred analyte, when present. The data summarized in Table 2 were obtained after correction for the daily recovery factor. Detailed results are listed in Appendix A. Glucuronides were never found. To the best of our knowledge, for HT and T the only documented sulfation site is the phenolic hydroxyl group; however, sulfation at the aliphatic -OH cannot be ruled out [36]. Consequently, HT-S was included among the analytes to be investigated, but it was never detected. Measurable concentrations of HT were determined only in three out of nineteen analyzed ewe cheeses in a range from 8 to 17 µg/kg dw (dry weight). As shown in Table 2, in ewe cheese, T, T-S, HT-3-S and HT-4-S were in the ranges < LOD-273 µg kg^−1^ dw, <LOD-33 µg/kg dw, 75–263 µg kg^−1^ dw and 91–816 µg kg^−1^ dw, respectively. 

In goat cheese, T, T-S, HT-3-S and HT-4-S were in the ranges <LOD-2296 µg kg^−1^ dw, 7.8–14 µg kg^−1^ dw, <LOD-23 µg kg^−1^ dw and 12–52 µg kg^−1^ dw, respectively, whereas in cow cheeses they were <LOD-1593 µg kg^−1^ dw, <LOD-30 µg/kg dw, <LOD-11 µg/kg dw and <LOD-27 µg/kg dw, respectively (Table 2). Ewe cheese showed significantly higher concentrations of polyphenols (sum) and both sulfate metabolites of HT than cow cheese (*p* < 0.05), while no significant differences were found for goat cheese, probably due to the low sample size.

## 3. Discussion

Concentrations of HT, T and metabolites in the thirty-six commercial cheeses were low, from few µg kg^−1^ (on dry matter basis) to some thousands, but higher than values measured in tissues of animals receiving feed enriched with olive oil by-products [6,8]. This was due to the polar nature of analytes, which determines their preferential excretion in body fluids as milk [7]. With regard to the parent compounds, HT and T, this latter was frequently detected, whereas HT was measured only in three ewe cheeses in the range from 9 µg kg^−1^ dw to 17 µg kg^−1^ dw. The presence of HT and T and their various glycosylated forms (ligstroside, nuzhenide, salidroside, verbascoside, etc.) has been mainly studied in olive oil, leaves and fruits, but these compounds are also widely diffused in plants that do not belong to *Oleacee* family [37,38,39,40]. In 2018, Bernardi et al. studied the phenolic profile of four maize cultivars estimating concentrations of HT from about 300 to about 800 mg kg^−1^, depending on maize cultivar [41]. Maize is one of the major ingredients of animal feed as well as barley and soy; however, for these latter vegetables specific analytical studies about the polyphenols involved in this research are not available. Accordingly, the compounds identified in the 36 commercial cheeses probably derive from vegetables used to produce animal feed or/and, for ewes and goats, from pasture grass. Ewe cheese was the richest in HT metabolites by a significant margin. High contents of T (>500 µg kg^−1^ dw) were measured in goat cheese and, sporadically, in cow cheese (Table 2 and Appendix A), as also shown by the representative chromatographic profiles in Figure 2. However, the difference among T concentrations by cheese were not statistically significant. Overall, cow cheeses had lower concentrations of analytes compared to those observed in ewe cheese, with a median value of polyphenol sum equal to 60 µg kg^−1^ dw and 455 µg kg^−1^ dw, respectively; in 6 of the 13 cow cheeses, the sum of polyphenols was even less than 50 µg kg^−1^ dw (Appendix A). On the other hand, the polyphenol sum in goat cheeses was not significantly different from that measured in the other two cheese types.

Some studies have been performed investigating milk and cheese total polyphenol contents, mainly measured as antioxidant activities, in relation to ruminant species and husbandry practices. In 2015, Velázquez Vázquez and coworkers measured total phenolic contents through antioxidant activity in ewe, goat and cow milk, finding that the concentrations in ewe milk were the highest, while those in cow milk the lowest [13]. With regard to husbandry practices and feeding habits, Hilario et al. (2010), Di Trana et al. (2015), Chávez-Servín et al. (2018) and Veskoukis et al. (2021) concluded that ewe and goat milk and cheese from animals eating fresh forage or freely grazing were richer of polyphenols than those from animals that consumed commercial feed confined to the barn [11,12,14,16]. All these studies were mainly based on data from spectrophotometric assays such as Folin-Ciocalteu, DPPH and ABTS. In reality, these kinds of tests not only cannot discriminate among various polyphenol sub-classes, but they measure generically sample antioxidant capacity, as demonstrated by Platzer and coworkers (2021) [42]. It is interesting to note that these results are compatible with our data, which showed higher concentrations of typical olive polyphenols in ewe cheese. As matter of fact, husbandry practices of small ruminants in central Italy (Appendix A) are extensive or semi-intensive and, therefore, farmed dairy ewes and goats always have full or partial access to free pasture, while dairy cows are generally kept in barns. However, our study is limited to only one polyphenol sub-class (phenyl alcohols) and, moreover, sample antioxidant capacity can be due to a plethora of substances. Since it is known that the antioxidant potential of milk and cheese depends on the animal diet, it is difficult to distinguish whether the observed differences are due to animal species (metabolism and species feeding habits) or to husbandry practices, which also influence diet (ingredients in the administered feed and access or not to pasture). Finally, research on humans and rats demonstrated that, among the phase II conjugate metabolites of HT, sulphates are preferentially formed with respect to glucuronidated metabolites of HT, as confirmed by our data [35,43,44]. On the contrary, our findings did not show a clear preference for the sulfation at position 3 with respect to position 4 (Table 2), whereas some authors have systematically observed higher concentrations of HT-3-S than HT-4-S in the urine of rats and human volunteers both before and after the administration of HT [32,33]. Since moisture percentage is a rough index of ripening time, statistical correlation between polyphenol concentration and moisture percentage was assessed, too. No significant correlation (Spearman’s rank correlation) was found and, therefore, ripening time does not seem play any role in polyphenol concentrations. No other remarkable correlations were detected.

## 4. Materials and Methods

### 4.1. Materials

Acetonitrile (ACN), methanol (MeOH) and ammonium acetate were purchased from Merck KGaA (Darmstadt, Germany). Formic acid was obtained from VWR Chemicals (Leuven, Belgium). Deionized water (HPLC grade) was obtained from a Millipore Q purification system (Mohlseim, France). SPE OASIS HLB Prime (150 mg/6 mL) and OASIS HLB (200 mg/6 mL) were supplied from Waters Corporation (Milford, MA, USA). Phree^TM^ Phospholipid Removal Tabbed 1 mL tubes were obtained from Phenomenex (Torrance, CA, USA). Tyrosol (T) was purchased from Merck KGaA (Darmstadt, Germany). Hydroxytyrosol (HT) was obtained from Alfachem s.r.l. (Milan, Italy). Hydroxytyrosol-sulphate (HT-S) was purchase from LGC Standards Ltd. (Teddington, UK). Tyrosol-sulphate (T-S), hydroxytyrosol-4-O-sulphate (HT-4-S), hydroxytyrosol-3-O-sulphate (HT-3-S), Tyrosol-glucuronide (T-G), hydroxytyrosol-3-O-glucuronide (HT-3-G) and hydroxytyrosol-d4 (HT-d4) were purchased from Toronto Research Chemicals (North York, ON, Canada).

### 4.2. Sample Preparation

One gram of cheese was extracted with 3 mL of a mixture of formic acid 0.5% in MeOH/water 80/20 (*v*/*v*). After shaking and centrifugation (4000 rpm, 10 min), the supernatant was transferred to −80 °C (30 min) for water freezing and then centrifuged (12,000 rpm at 0 °C, 10 min) and evaporated under nitrogen stream. In the meantime, the residual cheese was submitted to a second extraction with 3 mL of the same mixture and the supernatant freeze at −80 °C (30 min), and was then centrifuged and added to the previous supernatant portion for evaporation. After the dissolution of dry residue in 1 mL of ammonium acetate 100 mM (pH = 4), the sample was centrifuged (12,000 rpm at 0 °C, 10 min) and injected into the LC-Q-Orbitrap system. If necessary, an aliquot was diluted tenfold to quantify incurred polyphenols outside the working range.

### 4.3. LC-MS Conditions

Chromatographic separation was performed on a Thermo Ultimate 3000 High Performance Liquid Chromatography system (Thermo Scientific, San Jose, CA, USA). The analyte separation was carried out on a Acquity BEH C18 (2.1 × 150 mm, 1.7 µm, Waters, Milford, MA, USA) equipped with an Acquity guard column (2.1 × 5 mm, 1.7 µm, Waters, Milford, MA, USA). The flow rate and injection volume were 0.25 mL/min and 5 µL, respectively. Mobile phases were MeOH (A) and water with 2 mM ammonium acetate (B). LC gradient is reported in Appendix A. The run time was 22 min. The Q-Orbitrap mass analyser (Q-Exactive Plus, Thermo Scientific, San Jose, CA, USA) was equipped with heated electrospray ionization source (HESI-II). The acquisition was carried out in ESI-negative ionization mode performing full MS/dd-MS^2^ experiments. Optimized temperature of the source was 320 °C; capillary temperature was 280 °C; and the electrospray voltage was set at 2.5 kV. Sheath and auxiliary gases were set at 50 and 20 arbitrary units. Mass spectrometer was controlled using Excalibur 3.0 software and exact *m*/*z* of the compounds was determined using the Qualbrowser 2.0.3 tool of the software. The monitored precursor and fragment ions are detailed in Appendix A.

### 4.4. Validation Procedure

The developed method was validated according to Eurachem Guide [45]. Method linearity was evaluated in the range 0.5–1000 ng mL^−1^ by means of matrix-matched curves. Precision (repeatability, CV_r_; within-laboratory reproducibility, CV_wR_), recovery factors, detection limits (LODs) and quantification limits (LOQs) were estimated from the results of a spiking study. Cheese samples were spiked before extraction at four concentrations: 15, 80, 150 and 750 µg kg^−1^ dw for all the analytes, except for T which was added at concentrations ten-fold higher (150, 800, 1500 and 7500 µg kg^−1^ dw). Analytical sessions (four replicates) were repeated in three different days varying operator, calibration status of equipment and cheese type. Precision was assessed applying the Analysis of Variance (ANOVA) and recovery factors were calculated comparing the peak area found in the spiked samples against the relevant area in the matrix-matched calibration curve. Tyrosol-d4 was added prior to extraction in order to verify the extraction efficiency and not for quantitative purposes (recovery correction). In addition, tyrosol-d4 was used to compare relative retention times (RRTs). Internal quality control was carried out inserting two fortified samples into each analytical batch (positive controls); a daily matrix-matched curve was prepared for quantification purposes.

### 4.5. Analysis of Commercial Cheeses

Nineteen ewe, four goat and thirteen cow cheeses from various Italian regions and at different ripening stages were collected in local markets and analyzed to determine the contents of HT, T and some of their phase II metabolites. Moisture percentage was calculated from mass difference before and after 24 h of drying at 60 °C. Details about the collected cheeses are in Appendix A. Cheese classification was based on rheological properties [46].

### 4.6. Statistical Analyses

Polyphenol concentrations measured on commercial cheeses were not normally distributed (Shapiro–Wilk test) and differences among the three cheese types were analyzed by means of Kruskal–Wallis test followed by post hoc Dunn’s test. Spearman’s correlation was applied to assess the relationships among the measured parameters. Statistical analysis was performed by Stata package, version 16.1 (StataCorp LLC, College Station, TX, USA).

## 5. Conclusions

Levels of T, HT and their sulfate metabolites were measured in ewe, goat and cow cheeses produced in various Italian regions and at different stages of ripening. Ewe cheese showed higher contents of HT sulfate metabolites (HT-3-S and HT-4-S) as well as of polyphenol sum with respect to cow cheese. Polyphenol concentrations in goat cheese were not significantly different from those in the other two cheese types, but the small sample size may require further studies. In light of these findings, cheese cannot be considered a dietary source of these valuable polyphenols because HT and T health-promoting effects are observed with a daily intake in the order of milligrams and cheese portion size can ensure only a few micrograms per day. The developed method can be useful to analyze polyphenols in complex food of animal origin for which there are few validated procedures.

## Figures and Tables

**Figure 1 molecules-28-06204-f001:**
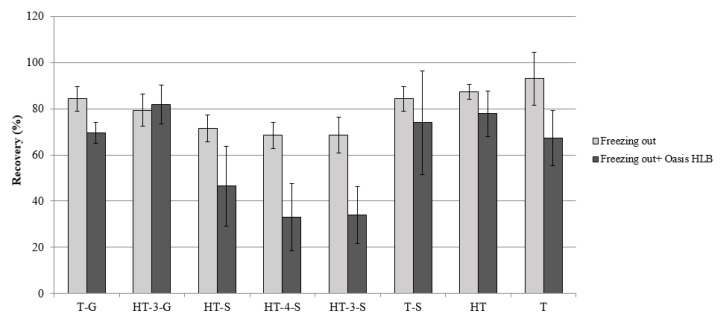
Recovery factors (mean of four replicates) with and without OASIS HLB clean up after the freezing out extraction. T-G: tyrosol-glucuronide, HT-3-G: hydroxytyrosol-3-glucuronide, HT-S: hydroxytyrosol-sulfate, HT-4-S: hydroxytyrosol-4-sulfate, HT-3-S: hydroxytyrosol-3-sulfate, T-S: tyrosol-sulfate, HT: hydroxytyrosol, T: tyrosol.

**Figure 2 molecules-28-06204-f002:**
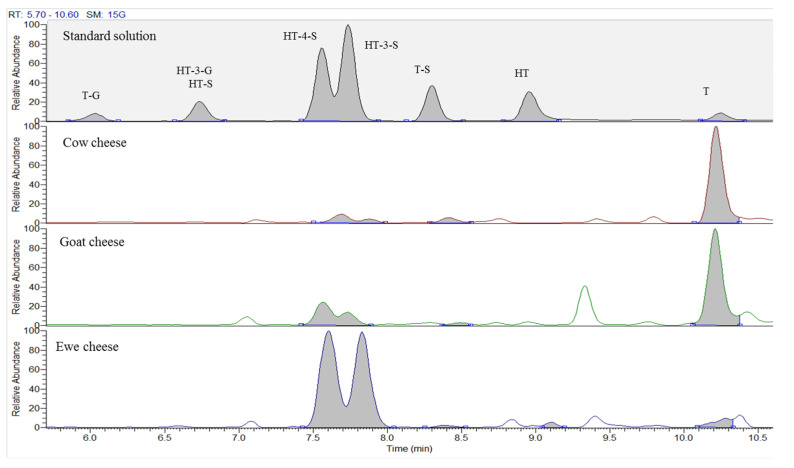
LC-HRMS/MS chromatograms of a standard solution of the eight analytes (50 ng mL^−1^) compared to chromatographic profiles of the three cheese types. T-G: tyrosol-glucuronide, HT-3-G: hydroxytyrosol-3-glucuronide, HT-S: hydroxytyrosol-sulfate, HT-4-S: hydroxytyrosol-4-sulfate, HT-3-S: hydroxytyrosol-3-sulfate, T-S: tyrosol-sulfate, HT: hydroxytyrosol, T: tyrosol.

**Table 1 molecules-28-06204-t001:** Results of validation study of the eight polyphenols in cheese.

Analyte ^1^	Spiking Concentration (µg/kg dw)	Recovery (%) ± SD (%)	CV_r_ (%) ^2^	CV_wR_ (%) ^2^
HT-3-G	15	73 ± 10	14	14
	80	71 ± 6	8.7	8.7
	150	75 ± 8	9.2	10
	750	77 ± 12	10	18
HT-S	15	63 ± 13	18	22
	80	65 ± 8	13	13
	150	62 ± 5	6.0	8
	750	62 ± 7	8.1	11
T-G	15	78 ± 13	13	17
	80	76 ± 9	12	12
	150	73 ± 7	8.3	9.1
	750	76 ± 9	8.5	13
HT-4-S	15	79 ± 6	7.7	7.7
	80	72 ± 7	8.5	9.4
	150	70 ± 6	6.2	8.9
	750	68 ± 6	8.5	8.5
HT-3-S	15	75 ± 5	5.2	7.2
	80	75 ± 7	9.8	9.8
	150	67 ± 4	5.5	6.9
	750	67 ± 8	7.1	13
T-S	15	81 ± 9	5.9	12
	80	81 ± 9	8.2	12
	150	75 ± 7	6.2	10
	750	66 ± 6	7.6	8.4
HT	15	86 ± 15	11	19
	80	89 ± 7	5.9	8.6
	150	81 ± 10	7.4	13
	750	72 ± 7	7.6	10
T	150	89 ± 14	8.7	17
	800	86 ± 6	7.4	6.5
	1500	81 ± 6	6.9	7.9
	7500	73 ± 5	6.2	7.8

^1^ HT-3-G: hydroxytyrosol-3-glucuronide, HT-S: hydroxytyrosol-sulfate, T-G: tyrosol-glucuronide, HT-4-S: hydroxytyrosol-4-sulfate, HT-3-S: hydroxytyrosol-3-sulfate, T-S: tyrosol-sulfate, HT: hydroxytyrosol, T: tyrosol; ^2^ CV_r_ and CV_wR_ are the coefficients of variation observed in repeatability and within-laboratory reproducibility conditions, respectively.

**Table 2 molecules-28-06204-t002:** Moisture (%) and concentrations (µg kg^−1^ dw) of hydroxytyrosol (HT), tyrosol (T) and their metabolites in cheeses.

Sample	Parameter ^1^	Frequency (%)	Mean	Median ^2^	Min	Max
Ewe cheese (19)	Moisture	-	33.25	36.06	17.66	40.19
	HT-4-S	100	279	213 ^a^	91	816
	HT-3-S	100	158	153 ^a^	75	263
	T-S	63	10	9	<LOD	33
	T	47	68	25	<LOD	273
	Sum	-	506	455 ^a^	224	1130
Goat cheese (4)	Moisture	-	54.51	57.24	38.05	65.50
	HT-4-S	100	28	24 ^ab^	12	52
	HT-3-S	75	12	11 ^ab^	<LOD	23
	T-S	100	11	11	7.8	14
	T	75	941	722	<LOD	2296
	Sum	-	985	776 ^ab^	40	2349
Cow cheese (13)	Moisture	-	34.44	33.83	23.49	45.73
	HT-4-S	46	7	<LOD ^b^	<LOD	27
	HT-3-S	15	<LOD	<LOD ^b^	<LOD	11
	T-S	77	14	12	<LOD	30
	T	54	277	60	<LOD	1593
	Sum	-	287	60 ^b^	<5	1621

^1^ HT-4-S: hydroxytyrosol-4-sulfate, HT-3-S: hydroxytyrosol-3-sulfate, T-S: tyrosol-sulfate, T: tyrosol ^2^ Different superscript letters (^a,b^) indicate significant difference among the three cheese types (*p* < 0.05).

## Data Availability

Not applicable.

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
