# Peer review of "Occurrence of Hydroxytyrosol, Tyrosol and Their Metabolites in Italian Cheese"

_molecules, 2023, doi:10.3390/molecules28176204_

Round 1
Reviewer 1 Report
Experimental article "Occurrence of hydroxytyrosol, tyrosol and their metabolites in Italian cheese" is devoted to the development of a method for assessing the content in cheeses made from various types of milk, substances such as hydroxytyrosol, tyrosol and their metabolites. According to scientific hypothesis, methods used, discussion of research progress and main conclusions this article is absolutely consistent with Molecules. The article discusses in detail the background and methodology for the determination of the above substances and sufficient evidence provided the possibility of using this technique in relation to cheeses. The article is very in detail in the methodological part with the presentation of a sufficient evidence base regarding the results obtained. At the same time, with such a detailed presentation experimental studies, the article does not fully reflect the relevance studies in terms of justifying the benefits of hydroxytyrosol, tyrosol and their metabolites for human when consuming products containing these substances. The article is written in clear language, detailed and well illustrated. It can be published after correcting some of the shortcomings.
Questions and remarks:
1. Already in the abstract it is indicated that this study showed that cheese cannot be considered an important dietary source of these valuable compounds - tyrosol (T) and hydroxytyrosol (HT). It would be appropriate to place information in the text of the article on how much of these substances can be considered significant in the product, so that this product can be considered as their source.
2. Lines 33-36 indicate that T and HT have a beneficial effect on health, but do not specifically describe what effect - it is recommended to cover this point in more detail.
3. Lines 37-40 indicate that the production of olive oil (which is rich in T and HT) generates a large amount of toxic waste, which negatively affects the environment, while the production of cheese produces no less waste - about 85% of the mass of raw materials (milk) and these wastes (whey) are also very harmful to the environment, this issue is not commented on in the article.
4. Lines 184-188 just say that low concentrations of T, HT and their derivatives were found in cheeses, but in animal tissues these concentrations were even lower. In this regard, it has been suggested that the main amount of these substances passes into milk. But since there are also few of them in cheeses, it may be advisable to provide data on the content of T, HT and their derivatives in the original types of milk and in the whey that is formed during the production of these cheeses. Perhaps just the secondary processing of whey will make it possible to obtain a product enriched with these substances.
5. Lines 205-207 say that it is possible that the difference in the amount of analytes in cheeses from different types of milk may be due to the different metabolism of animals. It would be logical to specify exactly which differences in metabolism affect the concentrations of these substances.
Author Response
Reviewer 1
Experimental article "Occurrence of hydroxytyrosol, tyrosol and their metabolites in Italian cheese" is devoted to the development of a method for assessing the content in cheeses made from various types of milk, substances such as hydroxytyrosol, tyrosol and their metabolites. According to scientific hypothesis, methods used, discussion of research progress and main conclusions this article is absolutely consistent with Molecules. The article discusses in detail the background and methodology for the determination of the above substances and sufficient evidence provided the possibility of using this technique in relation to cheeses. The article is very in detail in the methodological part with the presentation of a sufficient evidence base regarding the results obtained. At the same time, with such a detailed presentation experimental studies, the article does not fully reflect the relevance studies in terms of justifying the benefits of hydroxytyrosol, tyrosol and their metabolites for human when consuming products containing these substances. The article is written in clear language, detailed and well illustrated. It can be published after correcting some of the shortcomings.
Questions and remarks:
- Already in the abstract it is indicated that this study showed that cheese cannot be considered an important dietary source of these valuable compounds - tyrosol (T) and hydroxytyrosol (HT). It would be appropriate to place information in the text of the article on how much of these substances can be considered significant in the product, so that this product can be considered as their source.
Answer – Information about the minimum daily intake of T and HT to obtain beneficial effects on consumers has been added; in particular it is known the amount (at least 5 mg) to obtain antioxidant protection of blood lipids which is a property recognized by European Union (see Introduction and Conclusions).
- Lines 33-36 indicate that T and HT have a beneficial effect on health, but do not specifically describe what effect - it is recommended to cover this point in more detail.
Answer – More details on beneficial effects of HT and T have been added in the Introduction of revised manuscript adding the relevant literature.
- Lines 37-40 indicate that the production of olive oil (which is rich in T and HT) generates a large amount of toxic waste, which negatively affects the environment, while the production of cheese produces no less waste - about 85% of the mass of raw materials (milk) and these wastes (whey) are also very harmful to the environment, this issue is not commented on in the article.
Answer – The work does not focus on waste from the dairy industry, which does not have a major impact in Central Italy, where there is a large production of olive oil. The manuscript is just the latest work of our group resulting from a decade of studies on the possible re-use of olive oil by-products in animal feed. In addition, it is known that polyphenols do not interact with whey proteins of milk, but with casein (see below) and then they cannot be recovered from whey.
- Lines 184-188 just say that low concentrations of T, HT and their derivatives were found in cheeses, but in animal tissues these concentrations were even lower. In this regard, it has been suggested that the main amount of these substances passes into milk. But since there are also few of them in cheeses, it may be advisable to provide data on the content of T, HT and their derivatives in the original types of milk and in the whey that is formed during the production of these cheeses. Perhaps just the secondary processing of whey will make it possible to obtain a product enriched with these substances.
Answer - Thank you for this observation since this aspect must be better clarified in the manuscript. Our previous results (in farm experiment) showed that the concentrations of polyphenol compounds in sheep milk are similar to those in cheese, demonstrating that they are directly transferred from milk to cheese (Branciari et al., 2020, see "References" section, n.7). These data are coherent with literature evidences showing that polyphenols possess high affinity for milk casein proteins, but hardly interact with whey proteins (Han et al., 2019); therefore, in whey, polyphenols could not be recovered. In the "Introduction" section of revised manuscript this explanation was added and the paper of Han and coworkers cited in the "References" (n.9). Finally, the paper analyzes various commercial cheese products both to demonstrate method applicability and consumer assumption and, therefore, the original milk was not available.
- Lines 205-207 say that it is possible that the difference in the amount of analytes in cheeses from different types of milk may be due to the different metabolism of animals. It would be logical to specify exactly which differences in metabolism affect the concentrations of these substances
Answer – Thank you for this observation. In the revised manuscript this aspect has been clarified adding also studies about antioxidant capacity of milk and cheese in various species mainly relating antioxidant activity to feeding habits and to husbandry practices of animals. Although surely species metabolism affects polyphenol concentrations, it is not possible to establish an exact cause-effect relationship with available data. It has been demonstrated that the antioxidant potential of milk and cheese depends on the animal diet and feeding habits; therefore it is difficult to understand whether the differences observed among cheeses are due to animal species (metabolism and species-specific feeding habits) or to farming practices, which also influence the animal own diet (ingredients of administered feed and access or not to pasture).

Reviewer 2 Report
This study mainly measured hydroxytyrosol, tyrosol and their metabolites in different types of Italian cheese using validated analytical methods. The overall novelty and merit is low. This manuscript is more like a methodology paper, and only characterization data is presented. The authors also need to work more on the implications and interpretation of the results. Meanwhile, more meaningful data need to be generated. Therefore, the manuscript in this form is not recommended for publication.
Comments:
1. The introduction needs more info. Among all the polyphenols, why choose to study T and HT? What are their specific beneficial effects that general public would concern? Why is studying this important?
2. Section 2.1, the results relating to "OASIS HLB Prime and OASIS HLB cartridges" are confusing. When first introducing, the authors need to provide more information. What are these used for? From the results it seem that with the cartridges the recovery rates are even lower.
3. Figure legends need to be improved. What are these metabolites (T-S, T-G, etc) and why are they selected?
4. All tables are missing table legends.
5. Line 175, where is the correlation data? Also, the authors need to indicate why these specific metabolites are selected for analysis. Are they major metabolites? What are the correlations indicating and why are they important? There should be more interpretations of the results since there's not much data.
Some grammar issues are presented. Modification of language is needed.
Author Response
Reviewer 2
This study mainly measured hydroxytyrosol, tyrosol and their metabolites in different types of Italian cheese using validated analytical methods. The overall novelty and merit is low. This manuscript is more like a methodology paper, and only characterization data is presented. The authors also need to work more on the implications and interpretation of the results. Meanwhile, more meaningful data need to be generated. Therefore, the manuscript in this form is not recommended for publication.
Comments:
- The introduction needs more info. Among all the polyphenols, why choose to study T and HT? What are their specific beneficial effects that general public would concern? Why is studying this important?
Answer –HT and T are among the major polyphenols in olive products and they are even more predominant in olive by-products where hydrolysis processes transform the more complex polyphenols (olacein, oleochantal, verbascoside etc) to HT and T. In the “Introduction” of revised manuscript, the main beneficial effects of HT and T have been summarized to substantiate the interest about these bioactive molecules. In particular, their ability of protect blood lipids from oxidative processes is well known as such to be recognized also by European Union (Commission Regulation 432/2012). Moreover, as reported in the Introduction, the interest is also linked to high amounts of toxic wastes from olive oil industry produced in Mediterranean countries. Our study has been conceived in the context of re-use of by-products of olive oil mill in animal feed. Finally, from a methodological point of view, several papers report the measure of generic antioxidant activity of food by means of colorimetric tests (e.g. Folin-Cicalteau), wrongly associating this activity to polyphenol contents. The application of validated analytical methods that can distinguish individual polyphenolic substances can help investigate the real relationships between the antioxidant properties of foods and the presence of certain polyphenol molecules.
- Section 2.1, the results relating to "OASIS HLB Prime and OASIS HLB cartridges" are confusing. When first introducing, the authors need to provide more information. What are these used for? From the results it seem that with the cartridges the recovery rates are even lower.
Answer – Thank you for this observation. The manuscript has been revised to better explain this part. At the beginning of the development of sample treatment protocol we tried to insert also a clean-up step via SPE to minimise the injection of interfering substances. In literature the use of OASIS HLB and OASIS HLB Prime is sometimes reported as clean up in methods determining residues in cheese after freezing out extraction (see, for example, Xie et al. J. Chromatogr. B 1002, 2015, 19-29). Preliminary experiments showed that the three tested SPE cartridges (Phree, OASIS HLB and OSIS Prime) were not suitable to recover some of our analytes (Figure 1) and, accordingly, this step was not included in the final protocol.
- Figure legends need to be improved. What are these metabolites (T-S, T-G, etc) and why are they selected?
Answer – The figure legends have been improved in the revised manuscript as suggested by the reviewer.
- All tables are missing table legends.
Answer - Thank you very much for the suggestion, which improves manuscript readability. Legends have been added in the revised manuscript
- Line 175, where is the correlation data? Also, the authors need to indicate why these specific metabolites are selected for analysis. Are they major metabolites? What are the correlations indicating and why are they important? There should be more interpretations of the results since there's not much data.
Answer - This point probably was not sufficiently clear in the manuscript. The "correlation data" are data from statistical correlations (Sperman rank correlation) among all the measured parameters (moisture percentages and polyphenol concentrations) in the 36 Italian cheese samples collected from retail markets. Correlation analysis could gave information, for example, to assess whether moisture percentage was related to contents of some of determined polyphenols. Being moisture a rough index of cheese ripening time, a negative correlation could indicate degradation process during ripening. To avoid confusion this point has been better detailed in the revised text and statistical analysis of polyphenol data in the 36 collected cheeses has been refined using non parametric tests (non normal distribution).
Sulfates and glucuronates are the main metabolites described in literature in body fluids and tissues of humans and rats. In addition our previous researc measuring HT, T and metabolites by means of a separative technique, demonstrated the presence of sulphate metabolites (Branciari et al. 2017; Branciari et al. 2020; Branciari et al. 2021). Sulfates and glucuronides retain some of the properties of the parent compounds, contributing to the beneficial effects of HT and T and therefore it is appropriate tu measure them (see, e.g. the review of Serreli & Deiana. Biological relevance of extra virgin olive oil polyphenol metabolites. Antioxidants, 2018, 7(12): 170). Our investigations focused on representative phenolic compounds in olive oil characterized by high nutraceutical value and able to be transferred to milk and then to cheese.
As required, in the revised manuscript the interpretation of results has been enriched discussing further studies which relate husbandry practices to total polyphenol content (antioxidant capacity) of milk and cheese from milking animals belonging to different species (Hilario et al. 2010; Velázquez Vázquez et al. 2015; Di Trana et al. 2015; Chávez-Servín et al. 2018). Since these studies do not generally apply selective techniques such HPLC-DAD or, better, LC-MS, they cannot discriminate among the various sub-classes of polyphenols. Therefore our method can be successfully applied in studies to understand the role of HT and T in the improvement of cheese characteristics also in light of the reuse of olive oil by-products.